# Exploring Sirtuins: New Frontiers in Managing Heart Failure with Preserved Ejection Fraction

**DOI:** 10.3390/ijms25147740

**Published:** 2024-07-15

**Authors:** Ying Lu, Yongnan Li, Yixin Xie, Jiale Bu, Ruowen Yuan, Xiaowei Zhang

**Affiliations:** 1Department of Cardiology, Lanzhou University Second Hospital, Lanzhou 730031, China; ly19669669@163.com (Y.L.); yanxiwenny@foxmail.com (Y.X.); 15229899791@163.com (J.B.); lzyrw1723@126.com (R.Y.); 2Department of Cardiac Surgery, Lanzhou University Second Hospital, Lanzhou 730031, China; lyngyq2006@foxmail.com

**Keywords:** SIRTs, heart failure, heart failure with preserved ejection fraction, left ventricular diastolic dysfunction, development, treatment

## Abstract

With increasing research, the sirtuin (SIRT) protein family has become increasingly understood. Studies have demonstrated that SIRTs can aid in metabolism and affect various physiological processes, such as atherosclerosis, heart failure (HF), hypertension, type 2 diabetes, and other related disorders. Although the pathogenesis of HF with preserved ejection fraction (HFpEF) has not yet been clarified, SIRTs have a role in its development. Therefore, SIRTs may offer a fresh approach to the diagnosis, treatment, and prevention of HFpEF as a novel therapeutic intervention target.

## 1. Introduction

Heart failure (HF) is a complicated collection of clinical syndromes produced by aberrant alterations to cardiac structure and/or function due to several reasons that result in reduced ventricular systolic and/or diastolic function [1]. The European Society of Cardiology recommended in 2021 that HF be categorized into four categories: HF with reduced ejection fraction (HFrEF), HF with improved ejection fraction (HFimpEF), HF with mildly reduced ejection fraction (HFmrEF), and HF with preserved ejection fraction (HFpEF) [2]. HFpEF comprises approximately 50% of all cases of HF and is recognized as the most significant subtype, manifesting primarily as left ventricular diastolic dysfunction (LVDD); however, there is a lack of pertinent research on this topic [3,4,5,6,7,8,9,10]. The nicotinamide adenine dinucleotide (NAD)-dependent class III histone sirtuin (SIRT) family has recently been found to be closely associated with HFpEF development, with seven different isoforms of SIRT (SIRT1–7). It is involved in cell injury and maintenance of metabolic homeostasis through signaling pathways such as the eEF2K/eEF2 pathway, the 6 signaling pathway of the SIRT1/transmembrane BAX inhibitor motif, the Sirt3/MnSOD pathway, and the AMPK/PGC-1α pathway, causing endoplasmic reticulum stress, apoptosis, mitophagy, oxidative stress, and mitochondrial function [11]. This article reviews the current research on HFpEF and the molecular mechanisms of SIRTs in HFpEF development to provide new ideas for the diagnosis and treatment targets of HFpEF in the future.

## 2. HFpEF Overview

HFpEF refers to the clinical syndrome of pulmonary and systemic congestion caused by diastolic dysfunction and increased ventricular filling pressure due to slow and stiff left ventricular relaxation [11]. As medical staff further researched HFpEF and found that it involved lung, kidney, skeletal muscle, metabolism, and inflammation systems, it has been revealed that HFpEF is a syndrome involving multiple systems and multiple organ lesions. The clinical symptoms of early HFpEF are not obvious, and the traditional drugs for the treatment of HF have a poor prognosis for HFpEF, resulting in the failure of timely diagnosis and treatment of HFpEF and gradually increasing the incidence.

## 3. HFpEF Epidemiology

HFpEF has been shown to occur more frequently in older women with HF who are concurrently suffering from hypertension, elevated mean pulse pressure, obesity, atrial fibrillation, and anemia, as well as other comorbidities. According to the 2020 China Heart Failure Medical Quality Control Report, HFpEF accounts for approximately 38% of patients with HF in China [12]. As the population ages and hypertension, diabetes, obesity, and other diseases increase, HFpEF prevalence and mortality are rising. Epidemiology in western developed countries has shown that about 50% of hospitalized patients with HF are HFpEF/HFmrEF, and about 16% of outpatients with HF are HFpEF [6] in the China Heart Failure Center Database. The rehospitalization rate due to HF 1 year after discharge in 41,780 patients with HFpEF is 13.6%, and the mortality rate is 3.1% [13]. The main causes of death in patients with HFpEF in China are sudden death and myocardial infarction [14,15,16,17,18,19,20,21,22,23,24,25,26].

## 4. Sex Differences in HFpEF

In recent years, more and more studies have shown that there are significant sex differences in the occurrence and development of HFpEF [27,28,29]. In female patients, hormonal changes after menopause may affect the pathophysiological mechanisms of HFpEF. Compared to men, female HFpEF patients are more likely to exhibit features of metabolic syndrome, such as obesity, hypertension, and diabetes. Additionally, female HFpEF patients tend to have higher left ventricular mass indices and cardiac compliance, and experience more severe impairments in exercise tolerance and quality of life. These sex differences suggest that gender factors should be considered in the diagnosis, treatment, and management of HFpEF to develop more precise and individualized medical strategies. Therefore, future research should further explore the impact of sex on the pathogenesis of HFpEF and focus on gender-specific interventions in clinical practice.

## 5. Cardiac Structure, Function, and Vasculature

Healthy women and men have significant differences in left ventricular size and function. Even with the same body size, women have smaller left ventricular chambers and correspondingly lower left ventricular ejection volumes compared to men. With aging, diastolic function of the heart is also affected, and women have worse left ventricular diastolic function compared to men [30,31,32]. Additionally, both young and elderly women have smaller aortic root diameters, leading to increased pulse pressure. As women age, their arterial–ventricular coupling function weakens, making them more susceptible to HFpEF [27]. Furthermore, studies have shown that 75% of HFpEF patients have coronary microvascular dysfunction, and women have a higher prevalence of coronary artery disease. This not only increases systemic vascular resistance but can also lead to myocardial hypertrophy and fibrosis, resulting in diastolic dysfunction [33].

## 6. Comorbidities

Hypertension is more common in female HF patients than in males. Studies have shown that estrogen activates NO, causing vasodilation and reducing vascular stiffness. Additionally, hormones secreted by the ovaries reduce the activity of angiotensin-converting enzyme and renin. Therefore, after menopause, the decline in estrogen levels leads to increased vascular stiffness, doubling the risk of hypertension in women [34]. Surveys have shown that diabetes has a more significant impact on HF in women, increasing their risk by fivefold [35]. Recently, McHugh et al. [36] discovered factors such as sodium retention, volume overload, increased pro-inflammatory cytokines, and impaired cardiopulmonary function to explain the relationship between diabetes and HFpEF. Obesity and metabolic syndrome can cause diastolic dysfunction. Studies have found that postmenopausal women’s hearts are more sensitive to hypertension and obesity, exhibiting increased left ventricular mass and relative wall thickness [37].

## 7. Immune Function and Inflammation

Inflammation is considered one of the important pathophysiological changes in HFpEF. Numerous studies have shown that inflammation is associated with diastolic dysfunction and HFpEF. In HFpEF, comorbidities lead to systemic inflammation, reducing the protective effects of NO on the cardiovascular system, further causing myocardial cells to become stiff and hypertrophic [38]. Women have higher levels of pro-inflammatory genes and inflammatory cell gene expression than men, with higher activation levels of CD4 and CD8 cells, leading to increased systemic inflammatory responses [39]. Comorbidities such as diabetes and obesity can exacerbate inflammation. Additionally, most autoimmune diseases, such as rheumatoid arthritis and systemic lupus erythematosus, increase the body’s immune response. These autoimmune diseases have a higher prevalence in women and may be related to HFpEF [34].

## 8. Sex Hormones

Studies have found that a higher testosterone/estradiol ratio is associated with an increased risk of cardiovascular disease, coronary heart disease, and HF events. After menopause, the rapid decline in estradiol and sex hormone-binding globulin, coupled with the slower decline in testosterone, increases the risk of HF in postmenopausal women [40,41]. A follow-up survey [42] found that higher androgen levels were associated with greater increases in left ventricular mass, and women showed a higher mass-to-volume ratio (left ventricular concentric remodeling), which is a risk marker for HFpEF. Estradiol can counteract the concentric remodeling of the left ventricle induced by androgens. Therefore, the rapid decline in estradiol levels during menopause may be an important factor in the concentric remodeling of the left ventricle and increased risk of HFpEF in older women. At the same time, estrogen in women affects several molecular mechanisms that regulate ventricular diastole. Estrogen receptors not only regulate calcium influx through L-channels and sarcoplasmic reticulum activity but also protect the cardiovascular system by influencing NO synthase activity. Additionally, they act on cells and the nervous system to inhibit myocardial fibrosis [43]. Estrogen can also reduce catecholamine-induced vasoconstriction, promote vasodilation, and potentially increase the response of β2-adrenergic receptors [44]. However, after menopause, the decrease in estrogen levels gradually leads to the loss of cardiovascular protection and an increase in cardiovascular disease incidence, contributing to the higher prevalence of HFpEF in postmenopausal women compared to men.

## 9. Sex-Specific SIRT Mechanisms

According to the characteristics of women, there are specific SIRT mechanisms. Early studies on ovariectomized ApoE knockout mice found that vascular SIRT1 expression decreased, increasing endothelial dysfunction. Exogenous supplementation with 17β-estradiol restored SIRT1 expression and slowed vascular aging in mice, indicating that SIRT1 plays a crucial role in estrogen, protecting arteries from aging and atherosclerosis, thereby reducing the risk factors for HFpEF development [45]. Payavula et al. [46] found through a case-control study that VNTR polymorphism in intron 5 of the SIRT3 gene can serve as a molecular marker for detecting the onset of breast cancer, but further research is needed to study the prognostic and therapeutic significance of this SIRT3 polymorphism. Pro-inflammatory and pro-oxidative stress can stimulate myocardial fibrosis in HFpEF, while obesity can cause systemic inflammation, primarily involving monocytes and macrophages. Kolinko et al. [47] found through research that with increasing weight, SIRT1 gene expression levels increased. SIRT1 expression was highest in cells stimulated with IL-4 in obese subjects. This indicates that SIRT1 can promote the polarization of peripheral blood mononuclear cells by increasing STAT6 expression in overweight and low-risk obese young people. Studies that involved feeding high-sugar diets to male and female rats found that dietary fructose inhibited the expression of SIRT1 and insulin receptor substrate-2 mRNA in the aorta of female rats. Supplementing with resveratrol may restore fructose-induced metabolic and vascular dysfunction in both sexes through the production of eNOS and iNOS. Additionally, resveratrol may beneficially enhance SIRT1 and insulin receptor substrate-2 mRNA in females and insulin receptor substrate-1 mRNA in males [48]. HFpEF exhibits significant sex differences, not only in epidemiology but also in high-risk factors and treatments. Therefore, future research on HFpEF treatment should consider gender factors and explore personalized treatment plans.

## 10. Pathogenesis of HFpEF

The pathogenesis of HFpEF is complex and not yet clear, and current research focuses on LVDD and endothelial dysfunction. LVDD is considered to be the primary factor in the development of HFpEF; pathological hypertrophy is considered to be the main feature of HFpEF; about 30–60% of patients with HFpEF have left ventricular hypertrophy in clinical practice, and cardiac hypertrophy is mainly used in animal models of HFpEF to evaluate whether the model is successfully constructed [49,50,51,52]. It has been shown that in a mouse model of pathological cardiac hypertrophy, downregulation of peroxisome proliferator-activated receptor gamma coactivator 1-alpha (PGC-1α) following aberrant activation of the PI3K-AKT pathway in the heart caused a decrease in cardiac fatty acid oxidation and defects in mitochondrial function, leading to diastolic dysfunction [53]. Lopes et al. [54] found in their HL-1 cell experiments that treating cells with norepinephrine increased the expression of miR-145-5p. Additionally, the overexpression of miR-145-5p also upregulated the expression of ANP and BNP in cardiomyocytes. Moreover, in cells transfected with anti-miR-145-5p, the expression of norepinephrine-induced hypertrophy markers was significantly reduced. These data suggest that miR-145-5p may play a role in the pathophysiology of cardiomyocyte hypertrophy. From the pathophysiological perspective, myocardial fibrosis is the main cause of LVDD [55,56,57]. Dong et al. [58] established the HFpEF animal model and found that inhibition of miR-21 gene expression in fibroblasts led to a reduction in *Bcl-2* gene expression, a reduction in myofibroblasts, and an improvement in the degree of cardiac fibrosis. Additionally, conditions of pro-inflammatory and pro-oxidative stress also stimulate myocardial fiber metaplasia in HFpEF, and inflammation of cardiomyocytes promotes expression of the pro-fibrotic growth factor transforming growth factor-β, which stimulates fibroblast transformation into myofibroblasts while reducing matrix metalloproteinase-1 expression in the human HF, promoting cardiac fibrosis [59]. Inflammatory factors induce endothelial adhesion molecule expression, promote monocyte adhesion and infiltration, polarize macrophages in heart tissue, cause the secretion of inflammatory mediators, and induce aggravated myocardial fibrosis [60]. In addition, angiotensin II and aldosterone induce extracellular fibrosis by directly stimulating myofibroblasts to secrete collagen, activating NAD phosphate oxidase, and inhibiting matrix metalloproteinases to induce extracellular fibrosis [61].

Nitric oxide is considered an important factor in regulating endothelial function, leading to coronary microvascular endothelium inflammation under systemic inflammatory stress. Franssen et al. [62] found that in the myocardium of ZSF1-HFpEF rats, endothelial cells produce reactive oxygen species through the NO/cGMP/PKG signaling pathway during coronary microvascular endothelial oxidative stress, reducing myocardial nitric oxide availability. It results in decreased protein kinase G activity and promotes titin hypophosphorylation, thereby inducing endothelial dysfunction. In addition, endothelial cells can affect cardiomyocytes by releasing endothelin-1 when stimulated by inflammatory and mechanical factors, and in turn, cardiomyocytes can affect the coronary vasculature by releasing endothelin-1 and fibroblast growth factor 2. The effects between endothelial cells and cardiomyocytes imply that changes caused by endothelial dysfunction cause HFpEF [63]. LVDD and endothelial dysfunction are implicated in the pathophysiological mechanism of HFpEF and may serve as potential targets for future therapeutic interventions, according to the aforementioned findings. The complex pathogenesis mechanism of HFpEF necessitates additional investigation by researchers.

## 11. Application of Animal Models in HFpEF Research

HFpEF is a complex syndrome associated with multiple etiologies, exhibiting different phenotypic manifestations and involving various molecular mechanisms. Animal models can provide new insights into the underlying pathophysiological mechanisms and promote the development of effective therapeutic strategies. In recent years, the application of SIRT proteins in HFpEF animal models has gradually gained attention. Among these, HFpEF animal models include the “double hit” model and the “multiple hit” model.

In 2019, Schiattarella et al. [51] first reported that a high-fat diet combined with N-nitro-L-arginine methyl ester could successfully induce HFpEF in male 57 BL/6 N mice. Further mechanistic studies revealed that the iNOS-induced IRE1α-Xbp1s pathway dysregulation might be the pathogenic mechanism of HFpEF. In 2021, Deng et al. [64] established a “triple hit” HFpEF model of aging, high-fat diet, and mineralocorticoid using high-fat diet feeding combined with intraperitoneal injection of deoxycorticosterone pivalate (75 mg/kg). Further mechanistic studies confirmed that β-hydroxybutyrate could reduce NLPR3 inflammasome formation, decrease mitochondrial acetylation levels, alleviate mitochondrial dysfunction, and mitigate HFpEF myocardial fibrosis. In the same year, Withaar et al. [65] fed female C57BL/6J mice with HFD for 12 weeks, and during the 8th week, combined with the administration of angiotensin II [AngII, 1.25 mg/(kg·d)] via osmotic minipumps for 4 weeks, thus establishing a high-fat-aging-Ang II “triple hit” HFpEF model. This team further utilized this model to investigate the pharmacological value of liraglutide and dapagliflozin in HFpEF.

## 12. Research Findings on SIRT Proteins in Animal Models

JIA et al. [66] found that in a spontaneously hypertensive rat model treated with capsaicin, there was an increase in SIRT1 and glutamate decarboxylase protein expression and the number of positive cells in plasma, a reduction in reactive oxygen species production, and a decrease in the relative expression of proteins in the MAPKs pathway, thereby alleviating hypertension and cardiac hypertrophy. YOU et al. [67] found in a diabetic cardiomyopathy (DCM) model using db/db mice that in type 2 diabetes, miR-200a-3p interacts with FOXO3 to promote Mst1 expression and reduce SIRT3 and AMPK expression, thereby improving diabetes-induced cardiac dysfunction, myocardial injury, myocardial fibrosis, and myocardial cell apoptosis. LI et al. [68] established a coronary atherosclerosis mouse model by feeding ApoE−/− mice a high-fat and high-sugar diet and found that SIRT1 can regulate the proliferation and migration of endothelial progenitor cells through the wnt/β-catenin/GSK3β signaling pathway, thereby alleviating damage in the coronary atherosclerosis mouse model. Pathological myocardial hypertrophy can further develop into HFpEF. PENG et al. [69] established a transverse aortic constriction animal model and administered LCZ696 (an angiotensin receptor–neprilysin inhibitor) to mice. They found that LCZ696 could improve oxidative stress and pressure overload-induced pathological cardiac remodeling by regulating the Sirt3/MnSOD pathway. ZHANG et al. [70] established a myocardial hypertrophy animal model through transverse aortic constriction (TAC) and administered the compound FTZ via gavage. They found that FTZ alleviated myocardial hypertrophy and protected the myocardium through the miR-214-SIRT3 pathway. Inflammation and oxidative stress in myocardial cells contribute to myocardial fibrosis. LI et al. [71] established a diabetic SD rat model by intraperitoneal injection of streptozotocin. They found that hydrogen sulfide injection could activate CSE and autophagy through the SIRT6/AMPK signaling pathway, inhibit myocardial cell aging, reduce myocardial collagen fiber deposition, and improve diabetic myocardial fibrosis.

## 13. Limitations and Future Prospects

Animal models are not only channels for understanding the pathogenesis of diseases; they also provide information on how related therapeutic methods or interventions affect disease progression, prognosis, and associated complications. In recent years, with the increasing prevalence of HFpEF and the current lack of HFpEF-targeted therapeutic drugs in clinical practice, elucidating its pathophysiological mechanisms plays a crucial role in future drug development and treatment of HFpEF. Since HFpEF is a highly heterogeneous and complex disease, there is currently no animal model that can completely simulate human HFpEF. The existing HFpEF animal models established by various studies only encapsulate some characteristics of HFpEF patients, significantly limiting researchers’ understanding of HFpEF mechanisms and the progress in developing new drugs.

Current research clearly indicates that SIRTs are involved in the pathophysiological mechanisms of HFpEF and that influencing SIRTs activity has a positive effect on HFpEF-targeted therapy or prevention. Therefore, experimenters need to select appropriate existing models based on experimental purposes for studying the pathogenesis and developing effective therapeutic drugs. By using different HFpEF animal models and creating more models that better fit clinical HFpEF, researchers can further explore the interactions and regulatory mechanisms of SIRTs in HFpEF. This will better promote the understanding of HFpEF mechanisms and the discovery of new treatment strategies, providing a new direction for the prevention and prognosis of HFpEF (Table 1).

## 14. Studies of SIRTs and HFpEF

SIRT was first identified in transcriptional silencing in *Saccharomyces cerevisiae* cells as early as the last century [72]. SIRT2 is a highly conserved protein that impacts gene stability [73]. The presence of the SIRT2 homologous family protein SIRT in mammals has been confirmed by numerous studies. SIRT is a class III histone deacetylase that includes seven isoforms. SIRT1, SIRT6, and SIRT7 are found primarily in the nucleus, whereas SIRT3, SIRT4, and SIRT5 are found in mitochondria. SIRT2 is primarily found in the cytoplasm. SIRTs ultimately function in cardiovascular diseases such as myocardial infarction, coronary atherosclerotic heart disease, and HF by regulating the eEF2K/eEF2 pathway, the 6 signaling pathway of SIRT1/transmembrane BAX inhibitor motif, the Sirt3/MnSOD pathway, and the AMPK/PGC-1α pathway, which are involved in maintaining metabolic homeostasis, inhibiting inflammation, counteracting apoptosis, and inhibiting oxidative stress-related processes [74]. In recent years, growing investigations have indicated that every isoform of SIRT can interact with HFpEF (Table 2).

## 15. Biological Functions of SIRTs

In recent years, many studies have confirmed that SIRTs are involved in various physiological and pathological biological processes, exerting extensive effects on the functions of organs and tissues. Phenotypically, the SIRTs protein family is associated with multiple molecular, cellular, and physiological characteristics, such as inflammatory responses, metabolism, oxidative stress, apoptosis, and autophagy. Functional experiments have also confirmed their involvement in mitophagy and mitochondrial biogenesis [75].

### 15.1. Role of SIRTs in Inflammatory Responses

Increased expression of the SIRT1 protein can reduce the acetylation of the nuclear factor κ b (NF-κB) p65 subunit, thereby inhibiting TNF-α-induced NF-κB transcriptional activation and reducing the secretion of the inflammatory mediator TNF-α in a SIRT1-dependent manner [76]. A recent study showed that SIRT6 inhibits inflammatory responses via the NF-κB pathway, downregulating the expression of inflammatory factors IL-6 and TNF-α [77]. Additionally, Kurundkar et al. [78,79] demonstrated that SIRT3 deficiency altered macrophages’ pro-inflammatory response to lipopolysaccharides, increasing TNF-α production. Some studies also indicated that SIRT3 inhibits inflammasomes and reduces oxidative stress by downregulating IL-1β and IL-18, showcasing anti-inflammatory functions [80]. SIRT3 KO mice showed significantly increased inflammatory cell infiltration [81,82].

### 15.2. Role of SIRTs in Metabolism

SIRT1 is a key positive regulator of systemic insulin sensitivity, regulating pancreatic insulin secretion. SIRT1 can regulate glucose metabolism by upregulating AMPK, and AMPK activation can improve glucose metabolism imbalance [83,84,85,86]. SIRT3 negatively regulates aerobic glycolysis by inhibiting hypoxia-inducible factor 1α (HIF-1α) [87]. In pancreatic β-cells, SIRT6 deacetylates FoxO1, subsequently increasing the expression of glucose-dependent transporter 2 to maintain the glucose-sensing capability of pancreatic β-cells and overall glucose tolerance [88].

### 15.3. Role of SIRTs in Oxidative Stress

Oxidative stress, considered a significant factor in cellular damage, is caused by excessive production of reactive oxygen species (ROS). SIRT1 participates in regulating AMPK and its related pathways. For instance, AMPK can be activated by its upstream regulator liver kinase B1, and the activated AMPK mitigates oxidative stress damage by promoting insulin sensitivity, fatty acid oxidation, and mitochondrial biogenesis to generate ATP [89]. SIRT3 improves mitochondrial function by deacetylating liver kinase B1 and activating AMPK, thereby reducing ROS and lipid peroxidation [90]. SIRT3 activates the expression of FoxO3, which increases the transcription of MnSOD and CAT, thereby eliminating ROS [91]. SIRT6 also promotes the expression of AMPK, thereby upregulating the expression of the antioxidant-encoding genes MnSOD and CAT to inhibit oxidative stress [92].

### 15.4. Role of SIRTs in Apoptosis

SIRT1 mediates apoptosis by deacetylating FoxO proteins such as FoxO1, and upregulation of SIRT1 can inhibit apoptosis through the FoxO1/β-catenin pathway [93,94]. Overexpression of SIRT2 induces apoptosis by upregulating cleaved caspase 3 and Bax and downregulating the anti-apoptotic protein Bcl-2 [95]. Additionally, some studies found that SIRT3 has anti-apoptotic effects. SIRT3 deficiency leads to significantly increased apoptosis in septic mice, with elevated Bax and caspase 3 mRNA levels and reduced Bcl-2 mRNA levels, and SIRT3 KO mice show significantly increased caspase 3 expression [96].

### 15.5. SIRT1

SIRT1, predominately located in the nucleus, is also the most extensively studied SIRT. Its biological function is related to energy metabolism, oxidative stress, and endoplasmic reticulum stress. It has been found that SIRT1 interacts with and deacetylates PPARγ coactivator-1α in interleukin (IL) 6-deficient mice, and PPARγ coactivator-1α induces gluconeogenic genes, activates PGC-1α, enhances mitosis, restores energy metabolism in the myocardium, and improves HF [97]. In addition, Packer [98] indicated that it reduces the activation of SIRT1/PGC-1α/fibroblast growth factor and adenosine monophosphate-activated protein kinase and the inhibition of autophagy in type 2 diabetes, further causing mitochondrial dysfunction and oxidative stress and leading to myocardial fibrosis. Corbi [99] found that exercise training with cardiac rehabilitation could increase SIRT1 activity and β-hydroxybutyrate levels in patients with HFpEF and decrease oxidative stress to improve HFpEF. He et al. [100] induced HFpEF with a high-salt diet model in the study and indicated that canagliflozin attenuates cardiac hypertrophy and fibrosis by activating p-adenosine 5′-monophosphate-activated protein kinase and SIRT1 to induce PGC-1α expression. Sankaralingam et al. [101] found decreased cardiac SIRT1 expression, increased forkhead box O-class 1 acetylation, and increased atrogin-1 expression in an obese mouse model of HF, thereby reducing body weight, relieving myocardial hypertrophy, and improving diastolic function in obese mice. Costantino et al. [102] found that in a mouse model of metabolic cardiomyopathy, rat SIRT1 (rSIRT1) was able to attenuate the expression of key inflammatory cytokines IL6, IL-1β, and tumor necrosis factor-alpha (TNFα), and chronic exogenous rSIRT1 supplementation improved left ventricular ejection fraction, shortening fraction, and diastolic function to preserve cardiac function by restoring SIRT1 levels in the heart. Conti [103] found in the study that, by measuring SIRT1 activity in peripheral blood mononuclear cells and angiotensin-converting enzyme 2 (ACE2) activity, TNFα, and brain natriuretic peptide levels in plasma, patients with HFpEF had lower active SIRT1 and ACE2 levels than those of patients with HFmrEF and HFrEF, which could help distinguish HFmrEF/HFrEF from HFpEF phenotypes. This gives us further insight into potential therapeutic targets for HFpEF.

### 15.6. SIRT3

SIRT3, enriched in mitochondria, is a potential target of cardiac fibrosis and HF. Su et al. [104] showed upregulated 4-hydroxynonenal levels, downregulated glutathione peroxidase 4 expression, and increased p53 acetylation, resulting in cardiac fibrosis, by specifically knocking down SIRT3 in mouse cardiomyocytes. He et al. [105] found by knocking down SIRT3 in endothelial cells that deletion of SIRT3 decreased the expression of glycolytic enzyme PFKFB3 and altered glycolytic metabolism in endothelial cells, causing endothelial cell/cardiomyocyte interaction disorder and coronary microvascular dysfunction [25,106,107,108]. About two-thirds of patients with HFpEF are female [109,110,111]. Zeng et al. [112] showed that endothelial-specific SIRT3 knockout female mice fed a high-fat diet exhibited impaired coronary flow reserve and more pronounced diastolic dysfunction, perhaps one of the reasons for the higher prevalence of HFpEF in elderly female patients. Meanwhile, other studies are comparing elderly women with younger women; macrophage and pro-inflammatory factor infiltration is significantly increased in the hearts of elderly women, and the expression of SIRT1 and SIRT3 in their hearts is also significantly decreased, but these changes are not observed in male hearts [113]. These studies suggest that SIRT3 may play a role in sex differences associated with aging.

### 15.7. SIRT6

SIRT6 is distributed in the nucleus. Nicotinamide adenine dinucleotide has been found to activate SIRT6 in neonatal rats, thereby affecting nicotinamide mononucleotide adenyltransferase to protect cardiomyocytes from angiotensin II-induced myocardium hypertrophy [114]. According to Maksin et al.’s [115] research, transgenic mice overexpressing SIRT6 activated the pAMPKα pathway in cardiomyocytes, showed increased B-cell lymphoma 2 protein levels, inhibited activated B-cell nuclear factor kappa light chain enhancers and ROS, decreased phosphokinase B protein levels during hypoxia, and increased cardiomyocyte survival after hypoxia by obstructing the necrosis/apoptosis pathway. According to the study by Sundaresan et al. [116], SIRT6 binds to and inhibits the promoter of genes involved in insulin-like growth factor (IGF) signaling. This is achieved through interactions with c-Jun and the deacetylated histone 3-Lys9 locus. SIRT6 deficiency in mice leads to hyperactivation of IGF-related signal transduction genes, which in turn affects hypertrophy and the progression of heart disease in healthy cardiomyocytes (Figure 1).

### 15.8. SIRT2, SIRT4, SIRT5, and SIRT7

Mostly found in the cytoplasm, SIRT2 can also shuttle to the nucleus. In recent years, it has been shown that SIRT2 can deacetylate hepatic kinase B1 and promote the activation of the AMP-activated protein kinase pathway, thereby inhibiting aging-related and/or angiotensin II-induced pathological cardiac hypertrophy [117]. In the studies of Katare et al. [118], overexpression of SIRT2 inhibited acetylation of p53 and thereby toll-like receptor 4 damage to cardiomyocytes.

SIRT4, SIRT5, and SIRT7 have received less research attention in HFpEF compared with SIRT1, SIRT3, and SIRT6. SIRT4 is located in mitochondria and can regulate the oxidation of fatty acids, and inhibiting or decreasing its expression can enhance mitochondrial function and increase hepatic fat oxidation [119]. SIRT5 localizes to the mitochondrial matrix and can deacetylate it by activating carbamyl phosphate synthase 1, converting ammonia to nontoxic urea to modulate the urea cycle and reduce oxidative stress [120]. SIRT7 is expressed in the nucleolus, and Vakhrusheva et al. [121] found that deletion of SIRT7 increased apoptosis in primary cardiomyocytes by approximately 200% by knocking out the *SIRT7* gene in mice, and SIRT7 interacts with p53 in cardiomyocytes and can deacetylate p53, reduce cardiomyocyte death, and increase antioxidant stress response, thereby reducing cardiac hypertrophy (Figure 2).

## 16. Research on Pseudomolecular Functions of SIRTs

SIRTs possess deacetylase activity and play a crucial role in regulating apoptosis. One study found that SIRT1 can reduce histone acetylation levels in the promoters of genes such as AR, BRCA1, ERS1, ERS2, EZH2, and EP300, ultimately affecting cancer cell apoptosis. It remains to be further explored whether SIRT1 is involved in cardiomyocyte apoptosis in conditions such as heart failure [122]. Additionally, several transcription factors downstream of SIRT1, such as p53, NF-κB, and FoxO, are closely related to apoptosis. Upstream of SIRT1, besides miRNAs, a novel fibroblast growth factor 1 variant can reduce p53 activity by upregulating SIRT1-mediated deacetylation of p53, thereby counteracting doxorubicin-induced apoptosis. Since doxorubicin can induce cardiomyocyte apoptosis, targeting this pathway may represent a new therapeutic approach for doxorubicin-induced heart failure. Moreover, some drugs, such as ginsenosides, have been shown to inhibit or activate SIRT1 to regulate the apoptosis process. These studies further suggest that SIRT1 could be a potential therapeutic target for apoptosis [123].

In hepatocellular carcinoma, SIRT3 can enhance apoptosis by overexpressing caspase 9 cleavage and depleting SIRT3 in cancer cells [124]. Conversely, some studies have found that SIRT3 has anti-apoptotic effects. SIRT3 deficiency leads to significantly increased apoptosis in septic mice, with elevated Bax and caspase 3 mRNA levels and reduced Bcl-2 mRNA levels, and SIRT3 KO mice show significantly increased caspase 3 expression [96]. Therefore, SIRT3 exhibits different pro-apoptotic and anti-apoptotic roles in various diseases, necessitating further research to explore these opposing effects.

An in vivo study showed that SIRT4 exacerbates Ang II-induced cardiac hypertrophy by overexpressing and inhibiting MnSOD activity, providing new directions and targets for treating cardiac hypertrophy [125]. Another study found that SIRT6 expression is significantly reduced in the hearts of patients with chronic heart failure and animal models of heart failure. Additionally, LI et al. [126] found that overexpression of SIRT6 increased the survival rate of patients with transverse aortic constriction-induced heart failure, potentially related to the upregulation of telomerase reverse transcriptase and telomere repeat-binding factor 1, offering new research directions for heart failure treatment.

Members of the SIRTs family play important roles in vivo. Various studies have revealed their pseudomolecular functions, and using the latest research techniques, such as omics, gene knockout, and protein knock-in, can help uncover the specific molecular mechanisms of SIRTs. This provides new perspectives and directions for understanding the pathogenesis and therapeutic targets of HFpEF.

## 17. Treatment of SIRTs and HFpEF

Most of the treatments for HFpEF are now aimed at improving clinical symptoms, improving myocardial compliance, and controlling heart rate and rhythm. This includes, for example, ACE inhibitors, angiotensin receptor antagonists, calcium channel blockers, and β receptor blockers, but these drugs do not fundamentally improve their clinical prognosis, and many drugs used to treat HF with reduced ejection fraction do not have sufficient clinical evidence to influence HFpEF. Therefore, the treatment needs to control HFpEF risk factors. Combined with the findings above, the pathophysiology and pathogenesis of SIRTs and HFpEF are closely related, and polyphenols resveratrol and indole-3-propionic acid have been studied deeply in recent years.

## 18. Resveratrol

Resveratrol is a natural polyphenol compound [127] that has various pharmacological effects such as anti-oxidation, anti-tumor, cardioprotection, anti-inflammation, and neuroprotection [128] and belongs to the SIRT1 activator. SIRT1 can prevent cardiac hypertrophy and myocardial fibrosis after being activated, while SIRT activators have not been found so far. Rimbaud et al. [129] found that resveratrol inhibited cardiac dysfunction in high-salt diet rats with hypertensive HF by activating SIRT1 to protect mitochondrial fatty acid oxidation and peroxisome proliferator-activated receptor (PPAR)-α expression. Meanwhile, Wang and Tong [130] found that oral administration of resveratrol increased monocyte SIRT1 levels, increased insulin sensitivity, and decreased blood glucose in patients with type 2 diabetes mellitus complicated with coronary heart disease. Another study showed that resveratrol inhibited insulin-mediated aberrant migration and proliferation of vascular smooth muscle cells by activating SIRT1 and downregulating the PI3K/AKT pathway [131]. In a recent study, resveratrol was shown to upregulate PPAR-δ expression in wild-type Ppard-wt mice fed a high-fat diet to activate SIRT1 and enhance endothelial function in obese mice [132]. Ji et al. [133] showed through studies using ApoE−/− mice and umbilical vein endothelial cells isolated from ApoE−/− mice that resveratrol acts against atherosclerosis by downregulating the PI3K/AKT/mTOR pathway. Chen et al. [134] also found resveratrol to stimulate SIRT to mimic calorie restriction compounds, activate SIRT3 in cardiac fibroblasts, and inhibit the transforming growth factor-β/Smad3 pathway to improve cardiac fibrosis and cardiac function in in vitro and in vivo experiments. It also reduces melatonin-induced oxidative stress by interacting with SIRT3-dependent pathways and ameliorates myocardial ischemia/reperfusion injury.

In addition to resveratrol, there are other activators. It has been found that quercetin has been shown to inhibit oxidative stress injury in human cardiomyocytes by regulating mitophagy and endoplasmic reticulum stress through the SIRT1/TMBIM6 pathway [135]. Another study showed that quercetin could improve cardiac function by modulating the AMPK/SIRT1/NF-κB signaling pathway and inhibiting oxidative stress and inflammatory response [136]. 1,4-dihydropyridine derivatives, non-specific activators that effectively activate SIRT1, SIRT2, and SIRT3, can slow down the aging rate of C2C12 myoblasts in mice and increase mitochondrial density [137], which in turn affects cardiac function.

## 19. Indole-3-Propionic Acid

Indole-3-propionic acid is a metabolite produced by the gut microbiota that is involved in diastolic dysfunction, metabolic remodeling, oxidative stress, inflammation, gut dysbacteriosis, and intestinal epithelial barrier injury. One study found that indole-3-propionic acid reduced oxidative stress, inflammatory responses, and apoptosis in cardiomyocytes by inhibiting HDAC6/NOX2 signaling through the establishment of a mouse model of HF [138]. Li et al. [139] generated a mouse model of HFpEF and found that dietary supplementation with indole-3-propionic acid suppressed nicotinamide N-methyltransferase expression in a mouse model of HFpEF, and indole-3-propionic acid promoted SIRT3 expression by binding to arene hydrocarbon receptors, thereby reducing diastolic dysfunction in HFpEF. Perhaps therapeutic management of HFpEF can be achieved by altering the gut microbial composition or supplementing an indole-3-propionic acid diet.

In addition, SIRT3-AMPK signaling in rat and human skeletal muscle can be activated by oral nitrite, and to some extent, metabolic syndrome and cardiopulmonary hemodynamic disturbances can be ameliorated in the PH-HFpEF rat model [140,141]. In addition, Matsushimam et al. [142] found that treatment of HFpEF by increasing NAD levels increases SIRT3 activity and attenuates hyperacetylation of related mitochondrial enzymes.

To date, there are no specific treatment options that can reverse the development and outcome of HFpEF, but the current understanding of SIRTs in the treatment of HFpEF is still very limited and needs to be explored and studied through many basic and clinical trials (Table 3).

## 20. Comparison of SIRTs Agonists and Traditional Drugs

Novel energy metabolism drugs targeting myocardial mitochondria, specifically Sirtuins agonists (represented by NAD+), have gradually emerged as new treatments for heart failure. Traditional medications like angiotensin receptor–neprilysin inhibitors, sodium–glucose co-transporter 2 inhibitors, ACEI/ARB, β-blockers, and aldosterone receptor antagonists have limitations in treating heart failure. They cannot address issues such as myocardial energy metabolism, cardiomyocyte acetylation, and ventricular remodeling, resulting in unimproved patient prognosis and quality of life.

Sirtuins agonists, represented by NAD+ (coenzyme I), not only participate in material metabolism and energy synthesis but also serve as substrates to maintain the characteristics of poly(ADP-ribose) polymerase, cyclic ADP-ribose synthase, and histone deacetylases. Through this mechanism, they target and regulate histone deacetylation. By modulating metabolism, maintaining redox homeostasis, and regulating immune responses, Sirtuins improve heart failure symptoms and prognosis.

In one study, mitochondrial respiration in peripheral blood mononuclear cells (PBMCs) was compared between 19 hospitalized HF patients and 19 healthy participants. By isolating mitochondria damage-associated molecular patterns from human heart tissue and treating PBMCs, a sterile inflammation model was established in vitro. The study further found a causal relationship between systemic inflammation in HF patients and PBMC mitochondrial function. NAD+ can reduce systemic inflammation by inhibiting the pro-inflammatory activation of circulating immune cells. Increasing NAD levels may improve mitochondrial respiration and reduce pro-inflammatory activation in PBMCs of HF patients. Although this study lacked human efficacy data, the combined effects of NAD supplementation in maintaining heart function and reducing systemic inflammation provide a molecular basis for novel heart failure treatments [143].

Resveratrol, a selective SIRT1 agonist, has led to the development of synthetic SIRT1 activators (STACs) with improved potency and pharmacological properties. SRT2104 is the first SIRT1 STAC considered to have translational clinical potential. In phase I clinical trials, SRT2104 demonstrated good tolerance in healthy volunteers but had poor oral bioavailability [144]. Consistent with SIRT1 activation, despite its low bioavailability, SRT2104 reduced serum cholesterol and triglyceride levels and increased the HDL/LDL ratio in elderly patients after one month of administration [145]. Clinical studies on SRT2104 have also been conducted for cardiovascular function [146], type 2 diabetes [147], and other inflammatory diseases. Due to poor bioavailability [144], these subsequent clinical trials showed inter-subject variability in drug exposure and mixed results [138,148,149,150]. Therefore, once their pharmacological properties are optimized, SRT2104 and similar SIRT1 STACs may show greater therapeutic effects in age-related diseases. This comparison demonstrates that SIRTs agonists possess unique selectivity and targeting capabilities compared to traditional therapies. They also provide a new foundation for HFpEF treatment, and further in-depth research may offer new diagnostic and therapeutic drugs for HFpEF patients, improving long-term prognosis.

## 21. Clinical Trial Data

In the research of HFpEF, SIRT proteins have garnered widespread attention as potential therapeutic targets. Currently, there are several clinical trials focusing on the role of SIRT proteins in the development and progression of HFpEF. Early studies, using a randomized double-blind crossover design, found that a group of healthy obese men took resveratrol (150 mg per day) or placebo for 30 days. The men receiving resveratrol treatment showed significant reductions in resting metabolic rate, systolic blood pressure, and improvement in HOMA index. Muscle samples from these patients showed increased expression of SIRT1 and PGC1α proteins, increased AMPK activity, mitochondrial respiration, and fatty acid oxidation, demonstrating that 30 days of resveratrol supplementation induces metabolic changes in obese individuals, mimicking the effects of calorie restriction [151]. A prospective longitudinal observational study on patients undergoing continuous cardiac rehabilitation found that in elderly HFpEF patients after cardiac rehabilitation training, SIRT1 levels and catalase activity in peripheral blood mononuclear cells increased [152]. Recently, LU et al. [153] collected serum indicators of SIRT1, related inflammatory factors, and oxidative stress factors from HFpEF patients over the past three years. They found that serum SIRT1 levels were lower in the HFpEF group. Further analysis showed a negative correlation between SIRT1 and interleukin-6, tumor necrosis factor-α, C-reactive protein, malondialdehyde, and advanced oxidation protein product levels, suggesting that SIRT1 deficiency may induce oxidative stress and inflammatory responses, leading to the development and progression of HFpEF. Regression analysis results suggest that N-terminal pro-B-type natriuretic peptide (NT-proBNP) is closely related to poor prognosis in HFpEF patients. This provides a reference for the clinical diagnosis, treatment, and prognosis evaluation of HFpEF patients. HFpEF is a complex disease with increasing prevalence, requiring researchers to conduct more clinical studies to explore the relationship between HFpEF and SIRT, thereby further exploring new directions for the treatment and prognosis of HFpEF.

## 22. Conclusions and Outlook

The prevention and treatment of HFpEF, which is a complex clinical syndrome for which no efficacious drugs have yet been developed to reduce morbidity and mortality in patients, must be discussed in a step-by-step fashion. The current study clearly shows that SIRTs are involved in the pathophysiological mechanism of HFpEF pathogenesis and affect SIRT activity, having a positive effect on HFpEF-targeted therapy or prevention. SIRT activation generally influences cardiovascular disease, and SIRT level imbalances increase the risk of metabolic disorders, atherosclerosis, ischemic cardiomyopathy, and hypertrophic cardiomyopathy. Thus far, little evidence has substantiated SIRT1’s protective effects on the heart against hypertrophic stimulation, oxidative stress, ROS buildup, and apoptotic damage. SIRT1 also contributes to cardioprotection by controlling autophagy. SIRT3 overexpresses to protect mitochondrial function and reduce HF. SIRT6 activation might be a useful therapy option for atherosclerosis. SIRT5 and SIRT7 also have beneficial impacts. Many basic and clinical studies are required for the application and treatment of SIRT proteins in HFpEF, as well as the interaction, influence, and regulation between different types of SIRTs and effective pharmacological regulation, to be used for the prevention and treatment of clinical cardiovascular diseases and provide a new direction for the prevention, diagnosis, and treatment of HFpEF.

## Figures and Tables

**Figure 1 ijms-25-07740-f001:**
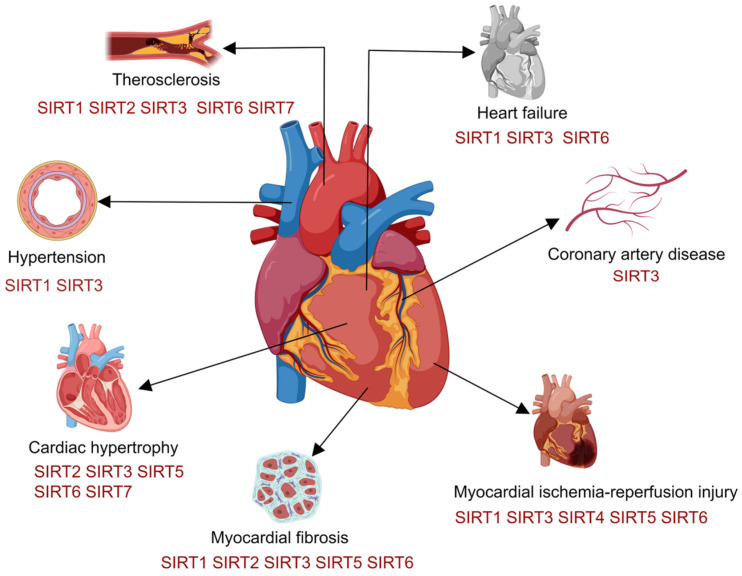
SIRT effects on the circulatory system. SIRT1, SIRT3, and SIRT6 have protective functions in cardiovascular disorders, including cardiac fibrosis, heart failure, atherosclerosis, and myocardial ischemia/reperfusion (MI/R) injury. In addition, SIRT2 is also protective against cardiac hypertrophy, myocardial fibrosis, and atherosclerosis. Furthermore, SIRT4 protects against atherosclerosis and MI/R injury. SIRT5 protects against heart hypertrophy, fibrosis, and MI/R injury. Finally, SIRT7 has been shown to protect against cardiac hypertrophy and atherosclerosis.

**Figure 2 ijms-25-07740-f002:**
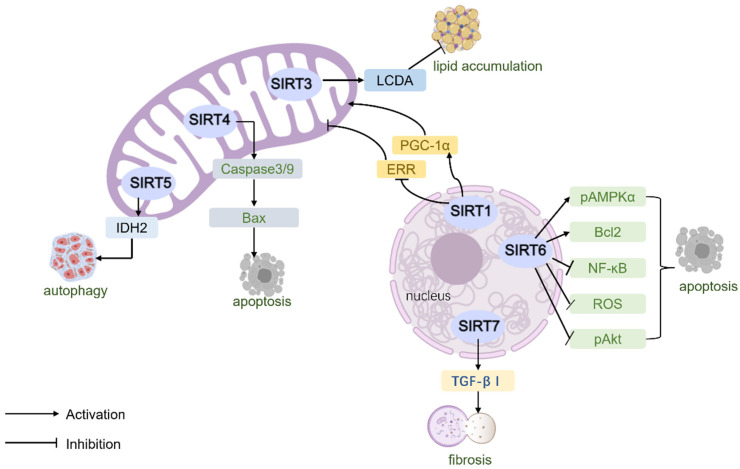
The role of SIRTs in heart failure.

**Table 1 ijms-25-07740-t001:** Summary of animal models with preserved ejection fraction in heart failure.

Category	Animal Models	Changes in Cardiac Function	Pathological Changes
Hypertension	Aortic constriction model	Preserved LVEF Increased SBP Increased E/A Increased E/E′ Increased LVEDP	Cardiomyocyte hypertrophy and ventricular hypertrophy; pulmonary congestion; interstitial and perivascular fibrosis
	Angiotensin II model	Preserved LVEF Increased SBP Decreased E/A Increased E/E′ Increased LVEDP	Cardiomyocyte hypertrophy and ventricular hypertrophy; pulmonary congestion; reduced myocardial capillary density; interstitial and perivascular fibrosis
	Dahl salt-sensitive rat model	Preserved LVEF Increased SBP Decreased E/A Increased E/E′ Increased LVEDP	Cardiomyocyte hypertrophy and ventricular hypertrophy; pulmonary congestion; decreased myocardial capillary density; interstitial and perivascular fibrosis
	Spontaneously hypertensive rat model	Preserved LVEF Increased SBP Increased E/A Increased E/E′	Cardiomyocyte hypertrophy and ventricular hypertrophy; decreased myocardial capillary density; interstitial and perivascular fibrosis
Obesity and diabetes model	STZ rat model	Preserved LVEF Decreased E/A	Cardiomyocyte hypertrophy; interstitial and perivascular fibrosis; decreased myocardial capillary density
“Multiple-Hit” model	ZSF1-obese rat model	Preserved LVEF Increased SBP Increased E/E′	Cardiomyocyte hypertrophy and ventricular hypertrophy; pulmonary congestion; interstitial and perivascular fibrosis; decreased myocardial capillary density; pulmonary congestion
	“2-hit” model	Preserved LVEF Increased SBP Decreased E/A Increased E/E′ Increased LVEDP	Cardiomyocyte hypertrophy and ventricular hypertrophy; pulmonary congestion; interstitial and perivascular fibrosis; decreased myocardial capillary density
	“3-hit” model	Preserved LVEF Increased SBP	Cardiomyocyte hypertrophy and ventricular hypertrophy; pulmonary congestion; interstitial and perivascular fibrosis; decreased myocardial capillary density; pulmonary congestion

**Table 2 ijms-25-07740-t002:** SIRTs protein family.

SIRTs	Subcellular Localization	Cell Localization	Physiological Activity	Participate in Pathophysiological Processes
SIRT1	Nucleus	Glucose metabolism, DNA repair, fatty acid metabolism, cell differentiation	Deacetylase	Myocardial hypertrophy, apoptosis, myocardial ischemia-reperfusion, heart failure
SIRT2	Cytoplasm	Cell cycle, fat metabolism	Deacetylase	Myocardial hypertrophy and myocardial ischemia-reperfusion
SIRT3	Nucleus, Cytoplasm	Mitochondrial autophagy, ATP synthesis, urea cycle, oxidative stress	Deacetylase	Myocardial hypertrophy, heart failure, apoptosis
SIRT4	Mitochondria	Insulin secretion, fatty acid oxidation, DNA repair	Depolymerizing enzyme, de-glutarylation	Myocardial infarction, heart failure
SIRT5	Mitochondria	Urea cycle	Desglutaryl enzyme, desmalonidase	Cell apoptosis
SIRT6	Nucleus	DNA repair	Deacetylase	Heart failure, myocardial hypertrophy, myocardial ischemia-reperfusion
SIRT7	Nucleus	Cell cycle	Deacetylase	Cell apoptosis

**Table 3 ijms-25-07740-t003:** Summary of SIRTs-related agonists and inhibitors.

	SIRT1	SIRT2	SIRT3	SIRT4	SIRT5	SIRT6	SIRT7
Inhibitor	Nicotinamide Leucine Oxidized paeoniflorin Suramin AGK2 HR73 MC2141	Nicotinamide Quercetin Suramin AK-7 AGK2 EX-527 MC2141 MC2494	Nicotinamide Quercetin 3-TYP AGK2 EX-527	Quercetin EX-527	Nicotinamide Quercetin Basalazine Peptides and amino acid inhibitors Suramin EX-527	Nicotinamide Quinazolinedione compound EX527	
Agonist	Resveratrol Quercetin 1pyr4-dihydropyridine small molecules Purple riveting Isoglycyrrhizic acid SRT172 SRT2104	Resveratrol SRT1460 SRT1720 SRT2183	Resveratrol aconitine polydatin magnolol ginseng polyphenols Chrysophanol SRT1460 SRT2183		Resveratrol MC3138	Anthocyanins Fucoidan Fluvastatin UBCS039	Quercetin

Note: AGK2 is 2-amino-3-[5-(2,5-dipneumophenyl)-2-furyl]-N-5-quinolyl-2-acrylamide; MDL-800 is 2-{[(5-bromo-4-fluoro-2-methylphenyl) amino] phosphoryl}-5-{[(3,5-dichlorophenyl) sulfonyl] amino} methyl benzoate; SIRTs are silent information regulator families; 3-TYP is pyridine-3-acetylene; UBCS039 is 4-(pyridin-3-yl)-4,5-dihydropyrrolo [1,2-a] quinoxaline.

## Data Availability

Data sharing is not applicable to this article.

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
