# Peer review of "Exploring Sirtuins: New Frontiers in Managing Heart Failure with Preserved Ejection Fraction"

_ijms, 2024, doi:10.3390/ijms25147740_

Round 1
Reviewer 1 Report
Comments and Suggestions for Authors
The title of the review is not adequate to the message including in the manuscript and wide part of it is dedicated to the animal models of HFpEF generally.
Authors should change the title or reorganize the manuscript and emphasize a sirtuins’ role.
Many parts of the manuscript are without any references and on the other hand many times there is an information “Error! Reference source not found” – authors should revise very carefully whole references list.
Author Response
Comment 1: The title of the review is not adequate to the message including in the manuscript and wide part of it is dedicated to the animal models of HFpEF generally. Authors should change the title or reorganize the manuscript and emphasize a sirtuins’role.
Response 1: Thank you for reviewing our manuscript and providing valuable feedback. We noted your comment regarding the substantial portion of the manuscript dedicated to animal models of HFpEF. Based on your suggestion, we have deleted and streamlined this section in the revised manuscript. The specific changes have been marked in red (located on page 7, line 145 to page 9, line 179).
Comment 2: Many parts of the manuscript are without any references and on the other hand many times there is an information “Error! Reference source not found” – authors should revise very carefully whole references list.
Response 2: Thank you for your suggestions. We have carefully reviewed the issues you mentioned and corrected the errors. The references have been clearly marked in the manuscript.

Reviewer 2 Report
Comments and Suggestions for Authors
Reviewer Comments
1. How are we to know that SIRT would be better than an other iosoforms through immunostimulants and immunosupressors that are currently used to treat HF?
2. Ref 56 is for cells in culture. Is there a better ref for the cardiomyocytes being directly infected and lysed in vivo, considering that is what is trying to be referenced here?
3. The manuscript lacks any information on the putative molecular function of the SIRT.
4. However, since the function of SIRT proteins is not defined at a molecular level, the authors have to provide further information on the physiological role of this protein.
5. The authors report on SIRT-dependent alterations in cardiovascular disorders. This reviewer has difficulties in appreciating a consistent regulation of mitochondrial expression.
Author Response
Comment 1: How are we to know that SIRT would be better than an other iosoforms through immunostimulants and immunosupressors that are currently used to treat HF?
Response 1: Thank you for raising such a meaningful question. Regarding this issue, we conducted a further review and found some studies indicating that agonists can be more effective than traditional drugs. For instance, a clinical study found that using the SIRT1 agonist SRT2104 can lower serum cholesterol and triglyceride levels in elderly patients, thereby reducing the risk of heart disease. The revised content has been added in red text (located on page 23, line 376 to page 25, line 404). However, there are relatively few related studies, and corresponding clinical research is lacking. This also provides a new research direction for the treatment and prognosis of HFpEF, holding significant potential for future research.
Comment 2: Ref 56 is for cells in culture. Is there a better ref for the cardiomyocytes being directly infected and lysed in vivo, considering that is what is trying to be referenced here?
Response 2: Regarding the cultured cells mentioned in reference 56, we have carefully reviewed the article and, based on your suggestion, have cited other relevant references. The corresponding modifications have been made in the manuscript and marked in red (located on page 6, lines 120-124).
Comment 3: The manuscript lacks any information on the putative molecular function of the SIRT.
Response 3: Regarding the lack of presumed molecular functions of SIRT in the article, we have taken this issue very seriously. To address this, we conducted further research and have added the relevant content to the revised manuscript. These additions have been marked in red (located on page 19, lines 303-326).
Comment 4: However, since the function of SIRT proteins is not defined at a molecular level, the authors have to provide further information on the physiological role of this protein.
Response 4: We understand your concern about the insufficient description of the physiological roles of SIRT proteins in the article. We accept this criticism and have provided a more comprehensive explanation of the physiological functions of SIRT proteins in metabolic regulation, aging, oxidative stress, and other areas. The detailed explanations and additional information have been marked in red in the manuscript (located on page 14, lines 207-239).
Comment 5: The authors report on SIRT-dependent alterations in cardiovascular disorders. This reviewer has difficulties in appreciating a consistent regulation of mitochondrial expression.
Response 5: Thank you for your feedback regarding the issue of SIRT proteins regulating mitochondrial expression in cardiovascular diseases. We recognize that the regulation of mitochondrial expression by SIRT is complex and may vary depending on the specific cardiovascular disease and cellular environment. Among the seven SIRT isoforms (SIRT1-7), for example, SIRT1 can lead to mitochondrial dysfunction and oxidative stress by reducing the activation of SIRT1/PGC-1α/fibroblast growth factor and AMP-activated protein kinase, as well as inhibiting autophagy, resulting in coronary microvascular dysfunction, myocardial inflammation, and myocardial fibrosis, thereby promoting the development of HFpEF. SIRT4, located in the mitochondria, can regulate fatty acid oxidation, and inhibiting or reducing its expression can enhance mitochondrial function and increase hepatic fat oxidation. SIRT5, localized in the mitochondrial matrix, can activate carbamoyl phosphate synthetase 1 to deacetylate it, converting ammonia into non-toxic urea, regulating the urea cycle, and reducing oxidative stress. These SIRT proteins have been confirmed to directly affect mitochondrial function. In our manuscript, we have supplemented the known roles of SIRT isoforms in mitochondrial regulation. We hope this addition addresses your concerns and provides a clearer understanding of the continuous regulation of mitochondrial expression by SIRT proteins in cardiovascular diseases.
